# TPU-KNN
# K Nearest Neighbor Search at Peak FLOP/s

**Felix Chern** [*]
Google Research
fchern@google.com

**Blake Hechtman** [*]
Google Core ML
blakehechtman@google.com

**Andy Davis** [*]
Google Core ML
andydavis@google.com

**Ruiqi Guo**
Google Research
guorq@google.com

**David Majnemer**
Google Core ML
majnemer@google.com

**Sanjiv Kumar**
Google Research
sanjivk@google.com

## Abstract

This paper presents a novel nearest neighbor search algorithm achieving TPU (Google Tensor Processing Unit) peak performance, outperforming state-of-the-art GPU algorithms with similar level of recall. The design of the proposed algorithm is motivated by an accurate accelerator performance model that takes into account both the memory and instruction bottlenecks. Our algorithm comes with an analytical guarantee of recall in expectation and does not require maintaining sophisticated index data structure or tuning, making it suitable for applications with frequent updates. Our work is available in the open-source package of Jax and Tensorflow on TPU.

## 1 Introduction

The $K$-nearest neighbor ($K$-NN) search problem has a wide range of applications in machine learning and information retrieval systems, including image search (Jia et al., 2021; Babenko and Lempitsky, 2016), semantic textual retrieval (Liu et al., 2009; Cer et al., 2018), anomaly detection (Gu et al., 2019; Omar et al., 2013), recommendation systems (Sarwar et al., 2002; Zhao et al., 2019), as well as serving as a component for a downstream tasks (Borgeaud et al., 2021; Guu et al., 2020; Lindgren et al., 2021; Shazeer et al., 2017). Given a query, the objective of $K$-NN is to identify $K$ closest datapoints from a database of finite number of data points in a vector space. The main challenge of designing a good $K$-NN algorithm is to compute accurate $K$-NN results while being computationally efficient.

Solving the $K$-NN problem on accelerators has emerging interests from both the academia and the industry (Johnson et al., 2021; Shanbhag et al., 2018; Zhao et al., 2020). Many accelerators can deliver hundreds of Tera Floating Point Operations Per Seconds (TFLOPS) vital to the neighbor distance computation. However, utilizing accelerators in $K$-NN problems is not straightforward; multiple issues in data locality, memory bandwidth, and multiple types of hardware parallelism need to be carefully considered to achieve high utilization. In this paper we extend the *roofline performance model* (Williams et al., 2009) to quantify the hardware characteristics accurately. As a result, we designed a $K$-NN algorithm to reach peak performance by the precise modeling of the accelerators, and our TPU implementation aligned with our predicted performance.

The main contributions of this work are:

---

[*]Equal contributions.

- We extend the roofline model to address the operation throughput differences of the instructions, essential to the algorithm analysis in this paper.
- We design an approximate $K$-NN algorithm with recall and performance guarantees based on our proposed roofline model.
- We conduct experiments verifying our TPU implementation of the algorithm accurately aligned with the performance model and achieves state-of-the-art speed-recall trade-offs on standard nearest neighbor search benchmarks.

## 2 Preliminaries

This section covers the necessary notations to work with the nearest neighbor search problem. Given a matrix $\mathbf{A} \in \mathbb{R}^{M \times N}$, we let $a_{i,j}$ denote the item at the $i$th row and $j$th column of $\mathbf{A}$, and $\mathbf{a}_i$ denote the $i$th *row-vector* of $\mathbf{A}$. We use the matrix $\mathbf{X} \in \mathbb{R}^{N \times D}$ to abbreviate a set-representation of a database $\mathbf{X} = \{\mathbf{x}_i\}_{i=1,2,\ldots,N}$ with $N$ data points, where each data point $\mathbf{x}_i \in \mathbb{R}^D$ is a row vector of the matrix $\mathbf{X}$ in a $D$ dimensional vector space. The set and matrix representation of database $\mathbf{X}$ are used interchangeably in this paper.

The $K$ nearest neighbor search problem is stated as follows. Given a database $\mathbf{X} \in \mathbb{R}^{N \times D}$ and a query vector $\mathbf{q} \in \mathbb{R}^D$, find the subset $\mathbf{S}^* \subset \mathbf{X}$ collecting the $K$-closest data points to $\mathbf{q}$:

$$\mathbf{S_q}^* = K\text{-}\underset{\mathbf{x} \in \mathbf{X}}{\operatorname{argmin}} \, \mathcal{D}(\mathbf{q}, \mathbf{x}), \tag{1}$$

where $\mathcal{D}(\mathbf{x}, \mathbf{y})$ is a distance measure such as Euclidean distance $\mathcal{D}_{\ell^2}(\mathbf{x}, \mathbf{y}) := \|\mathbf{x} - \mathbf{y}\|_2$ or the cosine distance $\mathcal{D}_{\cos}(\mathbf{x}, \mathbf{y}) := 1 - \frac{\langle \mathbf{x}, \mathbf{y} \rangle}{\|\mathbf{x}\| \|\mathbf{y}\|}$. A related problem is the maximum inner product search (MIPS), where the goal is to find the data points that have the highest inner products with the query:

$$\mathbf{S_q}^* = K\text{-}\underset{\mathbf{x} \in \mathbf{X}}{\operatorname{argmax}} \, \langle \mathbf{q}, \mathbf{x} \rangle. \tag{2}$$

MIPS is equivalent to the cosine similarity search when all data points are $\ell^2$-normalized.

## 3 Related work

Exhaustively searching all pair-wise distances between the query and the entire database is compute-intensive and often infeasible on many platforms. Therefore, a problem extensively discussed in the literature (Wang et al., 2014, 2015) is to find *approximate nearest neighbors* (ANN) in exchange of speed. By convention, the quality of ANN is measured by

$$\text{Recall} := \frac{|\mathbf{S_q} \cap \mathbf{S_q}^*|}{|\mathbf{S_q}^*|}, \tag{3}$$

where $\mathbf{S_q} \subset \mathbf{X}$ denotes the set of data points retrieved by the search method.

**Compressed domain search**  One class of ANN approaches is to search on a lossy-compressed problem domain. These methods are composed in two steps: a) search on compressed representation[2] of the original problem to find a set of candidate data points, b) compute the distances between the query and the candidate data points to select the top-$K$ results. Since only a subset of data points requires the exact distance computation, the overall cost is reduced.

The two steps can be composed in arbitrary ways. Locality sensitive hashing (Andoni et al., 2015; Neyshabur and Srebro, 2015) applies search followed by scoring; tree-search (Muja and Lowe, 2014; Dasgupta and Freund, 2008) applies the two steps recursively; graph-search (Malkov and Yashunin, 2018) iterates between two steps until the stopping condition is met. And the inverted file (IVF)

---

[2]Here we mean data structures like tree, graph, locality sensitive hash etc.

method (Jegou et al., 2010; Babenko and Lempitsky, 2014; Baranchuk et al., 2018; Guo et al., 2020) search on subset of data points indexed by the k-means centroids.

We see that there are two major challenges with the compressed domain search:

- Fractional search has a poor cache reuse rate because the candidate data points for each query rarely overlaps. We show optimizing the cache usage has a huge headroom for accelerators in Section 4.2.

- Tweaking the speed-recall trade-off is data-dependent and non-trivial to tune. The key result of Beyer et al. (1999) states that the distance contrast of neighbors diminishes with increasing dimensionality (also known as the curse of high dimensionality). Furthermore, the key result of Rubinstein (2018) states that sub-linear time nearest neighbor search with high recall is impossible for Euclidean, Manhattan, or Hamming distance; otherwise, it contradicts the Strong Exponential Time Hypothesis (Impagliazzo and Paturi, 1999).

Our work takes an opposite approach to focus on machine efficiency with zero search space pruning. Moreover, since our method computes all the distances, it is immune to the curse of high dimensionality.

**Accelerators** In this paper, the phrase *accelerators* represents a class of specialized hardware to accelerate machine learning workloads. In particular, we are interested in the novel platforms that deliver high FLOP/s for distance computation, namely Google TPU V3, V4, Nvidia GPU V100, and A100 in our analysis and evaluation.

Modern accelerators have special computation units for matrix multiplication, providing a higher operation throughput over the regular coefficient-wise operations. The corresponding units are tensor cores in Nvidia GPUs (Markidis et al., 2018) and systolic arrays in Google TPUs (Jouppi et al., 2017; Norrie et al., 2021). Addressing these operation throughput differences is essential to our algorithm design.

While accelerators excel in parallelism, developing an efficient $K$-selection algorithm on accelerators is still an active research area (Monroe et al., 2011; Shanbhag et al., 2018; Johnson et al., 2021; Zhao et al., 2020). Accelerators with higher FLOP/s introduce a higher opportunity cost of computing the $K$-selection problem instead of the distance computation. The trend of the increasing FLOP/s in accelerators motivated us to optimize the FLOP/s usage by reducing the time required for computing $K$-selection.

## 4 Methodology

This section presents a performance model to identify non-trivial bottlenecks on multiple platforms and demonstrates some fundamental limits when designing algorithms for $K$-NN and related problems, and we see that the cache inefficiency of the compressed domain methods introduces a significant cost on accelerators.

We model the accelerator's runtime as executing a sequence of *computation kernels*, where each kernel is a compiled subroutine on the accelerator used by the main program on the CPU. A kernel may be composed of one or several high-level operators: Einsum, ReLU, ArgMax, etc., and each kernel can have different performance characteristics.

Given a sequence of kernels $k_i$, we let $W_i$ denotes the total amount of work and $P_i$ denotes the operational speed. Our goal is to estimate the total time of a program:

$$t = \sum_i \frac{W_i}{P_i}. \tag{4}$$

In the following example, we focus on the MIPS problem. Let $\mathbf{Q} \in \mathbb{R}^{M \times D}$ and $\mathbf{X} \in \mathbb{R}^{N \times D}$ denote the queries and the database, the runtime of a generic approximate-MIPS program can be modeled as

$$t = \frac{\lambda W_{\mathcal{D}}}{P} + \mathcal{O}(\text{Auxiliary}) \geq \frac{\lambda W_{\mathcal{D}}}{P}, \tag{5}$$

Table 1: Hardware specifications for the generalized roofline model

| Name | $\pi$ (TFLOP/s) | $\beta$ (GB/s) | $\gamma$ (TCOP/s) |
|---|---|---|---|
| GPU V100 | 125 | 900 | 15.7 |
| GPU A100 | 312 | 1555 | 19.5 |
| TPU V3 | 126 | 858 | 4.0 |
| TPU V4 | 274 | 1144 | 4.3 |

where $W_{\mathcal{D}}$ denotes the total FLOPs required for searching the entire database, and $\lambda$ denotes the *search fraction*. We note that $P$ varies by algorithm and platform. Traditionally, compressed domain search methods minimize $\lambda$ but sacrifice cache efficiency. Our method use an alternative route to optimize $P$ instead.

### 4.1 Instruction throughput-aware roofline model

This subsection describes how we model the kernel-dependent performance $P$ on multiple platforms with a small extension of the roofline model.

The *classic roofline model* (Williams et al., 2009) is a function of machine peak performance $\pi$ measured in FLOP/s, machine peak memory bandwidth $\beta$ measured in bytes/s, and arithmetic intensity $I_{\mathrm{MEM}}$ expressed as the ratio of floating-point operations performed to data movement (FLOP/byte). The model states the performance is bounded by $P \leq \min(\pi, \beta \times I_{\mathrm{MEM}})$.

We desire to model kernels that has a mixture of floating point operations accelerated by dedicated hardware as well as other coefficient-wise operations. The coefficient-wise operations are abbreviated as COPs. Almost every non matrix multiplication operations are COPs, including vectorized add, multiply, compare, conditional-move, etc. We use the symbol $\gamma$ for peak COP/s on platforms, and define the instruction throughput intensity $I_{\mathrm{COP}}$ as the ratio between the number FLOPs and the number of COPs performed in a kernel (FLOP/COP). The attainable performance of a kernel is bounded by:

$$P \leq \min \begin{cases} \pi \\ \beta \times I_{\mathrm{MEM}} \\ \gamma \times I_{\mathrm{COP}}. \end{cases} \tag{6}$$

The statement is self-explanatory because the inadequate resources impede the kernel throughput. Table 1 lists the properties of selected accelerators for our analysis[3]. The roofline model is commonly used in accelerator profiling tools but not as frequently discussed in algorithm designs. The following sections show how the model prevents pitfalls due to the hardware constraints.

### 4.2 The memory bandwidth bound

This subsection demonstrates how to evaluate if a kernel hits the memory bandwidth wall. We associate the distance computation with three levels of BLAS (Dongarra et al., 1990). Level 1 BLAS describes vector operations on non-consecutive memory access, such as computing distances while traversing through a graph. Level 2 BLAS represents scoring a query with consecutively stored data points. Level 3 BLAS expresses batched query-database distance computation, often used in brute-force scoring.

Compressed domain searches are either level 1 or 2 BLAS due to the cache inefficiency. It has $\mathcal{O}(1)$ memory arithmetic intensity because the number of FLOPs is proportion to the bytes read. Combining (5) and (6) we have the following remark:

**Remark 1.** *Distance computations in compressed domain searches are memory bandwidth bounded. In our model, the runtime is lower bounded by: $t \geq \mathcal{O}\left(\lambda W_{\mathcal{D}}/\beta\right)$.*

---

[3]Readers can find these numbers from the accelerators' specification sheets.

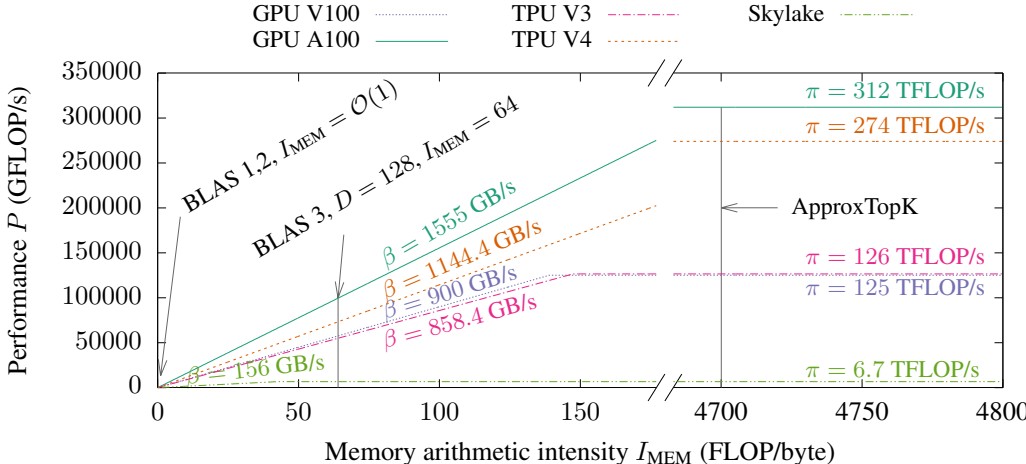

Figure 1: Memory rooflines of accelerators and a dual-sockets Intel skylake machine as a baseline. Each colored line denotes the maximum performance a platform could achieve, and each vertical line represents the memory arithmetic intensity of an algorithm. The intersections of the lines show the maximum performance of an algorithm could achieve on a platform. We label three levels of BLAS kernels and our algorithm described in Section 5.

To estimate the memory arithmetic intensity for level 3 BLAS, we continue to use $\mathbf{Q} \in \mathbb{R}^{M \times D}$ and $\mathbf{X} \in \mathbb{R}^{N \times D}$ for denoting queries and database. In many $K$-NN applications $N$ and $M$ are much greater than $D$. The corresponding memory arithmetic intensity is:

$$I_{\text{MEM}} = \frac{2MND}{4MN + o(MN)} \approx \frac{D}{2}. \tag{7}$$

The largest term in the denominator of (7) is the $4MN$ bytes of the query-database distances. We omit the insignificant terms and refer readers to (Golub and Van Loan, 2013, Section 1.5.4) for a comprehensive review on memory transfers in block matrix multiplications.

Figure 1 shows that the distance scoring kernels of different BLAS levels can easily hit the memory bandwidth wall. In order to attain high performance, we designed our algorithm to aggregate the results within the kernel to avoid writing the $\mathcal{O}(MN)$ bytes into memory.

### 4.3 The instruction bandwidth bound

The use of COPs (non matrix multiplication instructions) introduce another slowdown. We let $C$ denotes the number of COPs used per dot-product score in a kernel equipped with COPs and matrix multiplication instructions. There are $M \times N$ dot-product scores, so the total COPs used in a kernel is $CMN$. To prevent hitting the COPs bandwidth wall, we must satisfy:

$$I_{\text{COP}} = \frac{2\cancel{M}N D}{C\cancel{M}N} \geq \frac{\pi}{\gamma}, \tag{8}$$

$$\Rightarrow C \leq \frac{2D \times \gamma}{\pi}. \tag{9}$$

The number of COPs we can afford in the kernels is scarce. We take $D = 128$ as an example and substitute it into (9). We can only use 4 coefficient-wise instructions per dot-product for TPU V4, and 16 for GPU A100. We conclude with the following remark:

**Remark 2.** *Exact and generic $K$-selection algorithm cannot be efficiently implemented with the coefficient-wise operations for the selected platforms (GPU V100, A100, TPU V3 and V4).*

Because of Remark 2, we develop an approximate approach to achieve the peak performances.

# 5 Algorithm

---

**Algorithm 1:** PartialReduce for MIPS

---

**Input:** $\mathbf{Q} \in \mathbb{R}^{M \times D}$ Batch queries
**Input:** $\mathbf{X} \in \mathbb{R}^{N \times D}$ Database
**Input:** $2^W$ Bin size
**Output:** $\mathbf{V} \in \mathbb{R}^{M \times L}$ Top-$K$ values
**Output:** $\mathbf{A} \in \mathbb{N}^{M \times L}$ Top-$K$ indices

1 **for** $i \leftarrow 1$ **to** $M$ **do**
2 $\quad$ **for** $j \leftarrow 1$ **to** $N$ **do**
3 $\quad\quad$ $y_{i,j} \leftarrow \langle \mathbf{q}_i, \mathbf{x}_j \rangle$ ;
4 $\quad\quad$ $l \leftarrow \texttt{ShiftRight}(j, W)$ ; $\qquad\qquad$ /* Unrolled and does not cost COP */
5 $\quad\quad$ $b \leftarrow y_{i,j} > v_{i,l}$ ; $\qquad\qquad\qquad$ /* COP 1: Vectorized compare */
6 $\quad\quad$ $v_{i,l} \leftarrow$ **if** $b$ **then** $y_{i,j}$ **else** $v_{i,l}$ ; $\quad$ /* COP 2: Vectorized conditional move */
7 $\quad\quad$ $a_{i,l} \leftarrow$ **if** $b$ **then** $j$ **else** $a_{i,l}$ ; $\quad\quad$ /* COP 3: Vectorized conditional move */
8 $\quad$ **end**
9 **end**

---

Our algorithm consists of two kernels:

1. PartialReduce kernel computes the distances and partially aggregate the results from $M \times N$ distances to $M \times L$ distances with original indices.

2. ExactRescoring kernel is an *optional* kernel that aggregates the final top-$K$ results. The complexity is $\mathcal{O}(ML \log^2(L))$ by a bitonic sort followed by a truncation.

The PartialReduce kernel is where most of the time and compute takes place. See Algorithm 1 for an outline of the algorithm. We collect top-1 distances from the $L$ non-overlapping bins of size $2^W$ for each query, resulting high arithmetic intensities:

$$I_{\text{MEM}} \approx \mathcal{O}\left(\min\left(M, N\right)\right), \tag{10}$$

$$I_{\text{COP}} = \frac{2M\!N\!D}{CM\!N} = \frac{2D}{C}. \tag{11}$$

We show these arithmetic intensities can achieve high performance on real world database in section 6.1. See Appendix A.3 for the detailed expansion of the algorithm and how the arithmetic intensities are derived.

## 5.1 Recall estimation

This section shows the PartialReduce kernel can achieve high recall with good speed. We reformulate our problem in terms of balls and bins. We have $K$ balls representing the top-$K$ distances that are thrown into $L$ bins. The location of each ball is chosen independently and uniformly at random. We let $\mathbf{Z}$ denotes the random variable of the number of balls that do not have collisions. Following the recall definition (3) we have:

$$\text{Recall} \geq \frac{\mathbf{Z}}{K}, \tag{12}$$

which is a standard Birthday problem:

$$\mathbb{E}[\text{Recall}] \geq \frac{\mathbb{E}[\mathbf{Z}]}{K} = \left(\frac{L-1}{L}\right)^{K-1}. \tag{13}$$

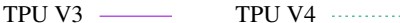

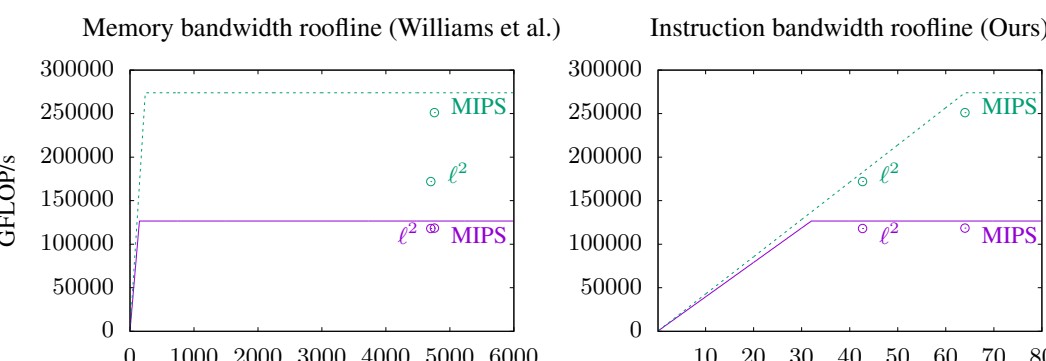

Figure 2: Roofline plots for MIPS and $\ell^2$ search benchmarks using the PartialReduce kernel. The colored lines denotes the attainable performance derived from Table 1. The figure on the left shows none of the benchmark is memory bandwidth limited. The figure on the right shows that our model gives a much tighter bound for $\ell^2$ on TPU V4. See also Appendix A.5 for detailed deviation of the numbers.

Our goal is to find the minimal $L$ such that the expected recall is greater equals to the target recall $r$. Finding $L$ is simple because (13) is invertible in the natural range $0 < r < 1$.

$$\mathbb{E}[\text{Recall}] \geq r \Rightarrow L \geq \frac{1}{1 - r^{1/(K-1)}} \approx \frac{K-1}{1-r}. \tag{14}$$

The approximation in (14) follows from Appendix A.4. Since $L$ is at the order of $K$, and in most applications $K \ll N$, the cost of the ExactRescoring kernel is amortized out. Thus we affirm the claim that our method attains high performance with an analytical recall guarantee. $\qquad\square$

## 6 Evaluation

In this section, we show that our proposed algorithm and implementation are near the hardware limit and lead to superior performance over the baselines of similar recalls. We applied our algorithm to two datasets from the public ANN benchmarks (Aumüller et al., 2020). In our first evaluation, we compare the measured FLOP/s to the theoretical peak governed by the proposed refinement of the roofline model (6), proclaiming our implementation is reaching the hardware peak performance. In the second benchmark, we compare the end-to-end performance with competitive baselines with pre-tuned parameters. We plot each algorithm's speed-recall curve and show ours achieves the state-of-the-art. Finally, we measure the algorithm's scalability by varying the dataset size and number of TPUs used.

### 6.1 Comparison with the theoretical peak

This section shows that our refined roofline model (6) captures additional performance characteristic over the classic roofline model, and demonstrates our kernels are having near optimal performances. We select the Glove[4] (Pennington et al., 2014) and Sift[5] (Jegou et al., 2010) datasets from the ANN benchmarks. Their corresponding distances are the cosine distance and the Euclidean distance. See the code snippets in Appendix A.1 and A.2.

---

[4]Released in Apache license 2.0.
[5]Released in CC0 public domain.

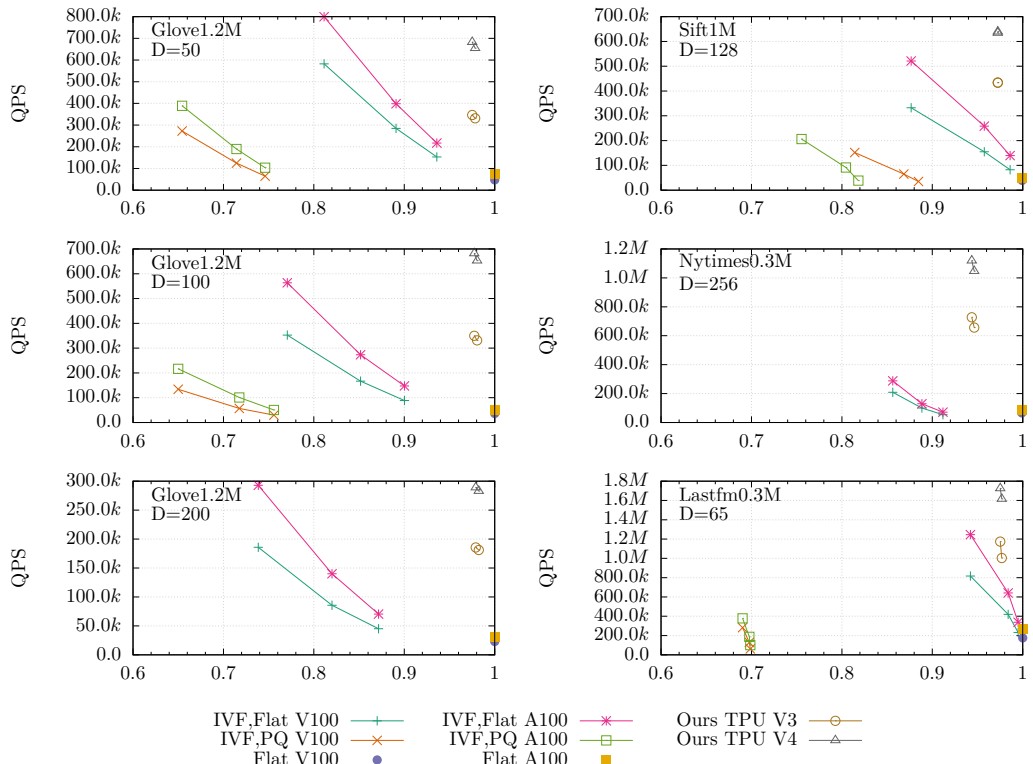

Figure 3: Recall-speed trade-off benchmarks. The x-axis is recall for $k = 10$; up and to the right the better of the trade-off. The GPU methods (IVF-Flat, IVF-QP, and Flat) are released by Faiss (Johnson et al., 2021). For each IVF* benchmark, the search fractions are $\lambda = \{0.24\%, 0.61\%, 1.22\%\}$.

See Figure 2, the colored lines represent machines' max performances, and the dots represent each benchmark with its measured FLOP/s. The classic roofline on the left shows that our in-cache aggregation strategy has a large memory arithmetic intensity ($\sim$4,700) exceeding the memory bandwidth ridge points $\pi/\beta$. However, it is difficult to diagnose why the Euclidean distance search does not perform well on TPU V4 from the classic roofline plot.

Fortunately, when combined with the instruction bandwidth roofline we can tell the performance regression is caused by hitting the coefficient-wise operation throughput wall. Therefore we affirms the claim that our MIPS solution is reaching the peak FLOP/s, and our Euclidean distance search solution is meeting the compute bound on TPU V4 and attaining the peak FLOP/s on TPU V3.

## 6.2 Recall-speed benchmark

To evaluate the effectiveness of the $K$-NN algorithm in a realistic setting, we adopted the methodology of public ANN benchmarks (Aumüller et al., 2020) to compare the end-to-end performance against other methods on the following datasets: Glove (Pennington et al., 2014), Sift (Jegou et al., 2010), NYTimes (Dua and Graff, 2017), and Last.fm (Bertin-Mahieux et al., 2011). The typical ANN benchmarks are only performed on a single platform. However, it is non-trivial to either port our TPU algorithm to GPU or vice versa. Alternatively, we selected the following GPUs with parity in peak performance to TPU (Table 1).

We select the Faiss GPU (Johnson et al., 2021) implementation as our baseline. Faiss provides three algorithms: Flat, IVF-Flat, and IVF-PQ. The Flat algorithm performs a brute-force search, and the IVF-Flat and IVF-PQ algorithms corresponds to the inverted file method with and without the product quantization (Jegou et al., 2010; Johnson et al., 2021). We use the repository's suggested inverted file size (16384) in the IVF methods.

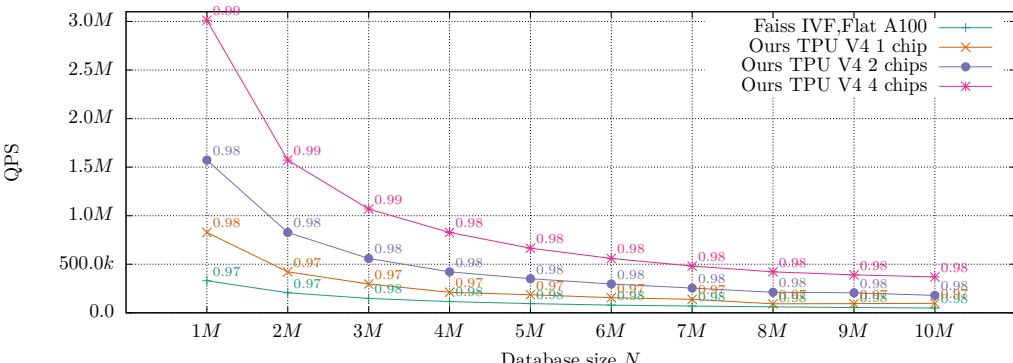

Figure 4: Scalability benchmark. The labeled numbers are the measured recalls. The multi-TPU implementation is listed in Appendix A.6.

Figure 3 shows our performance significantly outperforms competing methods in the high recall regions. We highlight that our method has a consistent recall-speed trade-off over different datasets, because our recall only rely on the order statistics instead of the information encoded in the compression domain search methods, which may vary by the datasets. Since our method scores all the pair-wise distances, our method is immune from the curse of high dimensionality.

## 6.3 Scalability benchmark

In the final benchmark, we examine the scalability of the algorithm from three aspects. First, we verify if the measured performance is inverse proportional to the database size. Second, we compare the scaling characteristics to the fastest GPU implementation. Last but not least, we are interested in knowing if our algorithm can horizontally scale by the number of TPUs.

We conduct our evaluation on TPU V4 and Nvidia GPU A100, which have similar peak performance and memory bandwidth. We sample the Yandex Deep dataset[6] (Babenko and Lempitsky, 2016) into ten different scales and measure the QPS of each approach with a similar recall. Figure 4 verifies all measurements align with the ideal scalability model: $QPS \propto \#chips/N$. Our method remains top performance on all database sizes and linearly scales with the number of TPU chips.

# 7 Discussion and future work

In Section 6, we benchmark our method against others on platforms with similar performances. Some questions might arise: "Is the performance gain an algorithmic optimization or due to platform efficiency?" "Can we achieve the same performance gain on GPU?" "The existence of efficient fractional-search on accelerators?" We address these questions in this section.

## 7.1 Platform discussions

We first discuss the modeling perspective of performance differences between platforms. In Section 4, we show that the memory bandwidth and instruction throughput bound applies to both GPU and TPU. For instance, it follows that to attain peak performance on every hardware platform, having the number of instructions used for collecting (approximate) top-k elements within $2\gamma \cdot D/\pi$ per distance computation is a *necessary condition*.

Although our Algorithm 1 is platform-independent, achieving the hardware peak performance requires many low level implementation details at the machine level, including cache management, preventing cross-core memory synchronization, in-register accumulation, and instruction scheduling. Typical high-performance libraries such as MKL, cuBLAS, and Google TPU compiler use platform-specific assembly to take full control of the stated requirements.

---

[6]Released in CC BY 4.0.

Nevertheless, we cannot use the high-level interface of these libraries, because Algorithm 1 only performs well when it is integrated into the inner loop of distance computations[7]. Moreover, these libraries are all close-sourced, thus increases the difficulty on the implementation.

Fortunately, we have the access to TPU compiler internals, and we have integrated Algorithm 1 into the compiler to generate the desired assembly code to solidify our analysis. Thus we leave implementations of other platforms to future works.

### 7.2 Algorithm discussions

The roofline complexity of the fractional search is identical to BLAS-2 (matrix-vector multiplication), which is memory bandwidth bound. When the cycles spend on data transfer are mutually exclusive to our method, it introduces an enormous opportunity cost. Nevertheless, we see an opportunity in a heterogeneous architecture because a fractional search on the host is *not* mutually exclusive to applying our method to accelerators.

A motivating example is the multi-billion nearest neighbor search, where fitting the dataset into device memory is possible (through device sharding, which TensorFlow and Jax have native support) but not economical. Since brute-force distance computations are often involved in the auxiliary data structures when performing the fractional search, we may replace the brute-force portion with TPU in conduction with the remaining search off-device. We note that heterogeneous architectures with off-device storage such as host-RAM or even SSD (Jayaram Subramanya et al., 2019; Ren et al., 2020; Chen et al., 2021) are great starting points for future research.

## 8 Conclusion

Accelerator-based machine learning has become the mainstream in academics and industries. However, the performance characteristics of accelerators are counter-intuitive and difficult to program. In this paper, we propose a roofline-based complexity analysis framework to discuss the optimality of the algorithms without low-level optimization details: unrolling factors, batch window sizes, vectorization, and systolic array scheduling, which are platform-dependent and lengthy to read. We demonstrated several examples of inferring the hardware performance limits by simply addressing the kernel's total FLOPs, byte transferred, and the number of coefficient-wise instructions used. Our refined model foreshadowed non-trivial performance regression caused by the coefficient-wise instructions bandwidth. We took it into account to design a new algorithm for $K$-NN and achieved peak performance on TPU. Finally, our experiments showed that our method outperformed state-of-the-art baselines on platforms with similar performance characteristics, which are known to be hard to beat.

## Acknowledgments and Disclosure of Funding

We would like to thank the XLA team for the continuous effort on developing the state-of-the-art compiler and the full support on enabling our new op: `approx_max_k`. We are also grateful to the Google ScaNN team for the joint effort on bridging the impactful $K$-NN problem into the accelerator ecosystem. Last but not least, we thank to Peter Hawkins, Edward Schwartz, and Mani Varadarajan for code reviews in Jax and Tensorflow, and Erik Lindgren for the proof reading of this paper.

This work was performed and funded by Google.

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
