# A   Appendix

## A.1   MIPS implementation

```
import jax
@jax.jit
def MIPS(query, database):
  scores = jax.numpy.einsum('ik,jk->ij', query, database)
  return jax.lax.approx_max_k(scores, k=10, recall_target=0.95)
```

Listing 1: Jax code for maximum inner product search (MIPS)

Listing 1 demonstrates a maximum inner product search (MIPS) kernel implemented with Jax. Tensorflow users can use the `tf.math.approx_max_k` interface; the underlying XLA compiler delivers the same kernel. There are several options to control the behavior of `approx_max_k`, listed as below:

1. `reduction_dimension` specifies the dimension in which to search. Default -1 (the last dimension.)

2. `recall_target` derives the number of bins $L$ of the PartialReduce kernel output. Default 0.95.

3. `reduction_input_size_override`. When set to a positive value, it overrides the size determined by input for evaluating the recall and bin numbers $L$. Users could use this option to control the kernel output size in the distributed environment.

4. `aggregate_to_topk`. When set to `True` emits the ExactRescoring kernel. Default: `True`.

We also provid a separated `approx_min_k` interface for finding minimum distances, which is used in the Euclidean distance search.

## A.2   Euclidean distance search implementation

```
@jax.jit
def l2nns(qy, db, db_half_sqnorm):
  dots = jax.numpy.einsum('ik,jk->ij', qy, db)
  dists = db_half_sqnorm - dots
  return jax.lax.approx_min_k(dists, k=10, recall_target=0.95)
```

Listing 2: Jax code for nearest neighbor search in the Euclidean space.

Listing 2 is the Jax implementation of Euclidean space nearest neighbor search. We made a few adjustments to speed up the computation. First, for every query vector $\mathbf{q}$, the following search produces the same result:

$$\mathbf{S}_{\ell^2}^* = K\text{-}\operatorname*{argmin}_{\mathbf{x} \in \mathbf{X}} \|\mathbf{q} - \mathbf{x}\|_2 \tag{15}$$

$$= K\text{-}\operatorname*{argmin}_{\mathbf{x} \in \mathbf{X}} \|\mathbf{q} - \mathbf{x}\|^2 \tag{16}$$

$$= K\text{-}\operatorname*{argmin}_{\mathbf{x} \in \mathbf{X}} \|\mathbf{q}\|^2 + \|\mathbf{x}\|^2 - 2\langle \mathbf{q}, \mathbf{x} \rangle \tag{17}$$

$$= K\text{-}\operatorname*{argmin}_{\mathbf{x} \in \mathbf{X}} \|\mathbf{x}\|^2 - 2\langle \mathbf{q}, \mathbf{x} \rangle \tag{18}$$

The last equation holds because omitting the query norm does not affect the rank of each result. Nevertheless, (18) still uses 2 COPs for the distance computation (one subtract and one multiplication). We can further reduce it to 1 COP by pre-computing the halved squared-norm:

$$\mathbf{S}_{\ell^2}^* = K\text{-}\underset{\mathbf{x}\in\mathbf{X}}{\arg\min}\ \frac{\|\mathbf{x}\|^2}{2} - \langle\mathbf{q}, \mathbf{x}\rangle \tag{19}$$

### A.3 MIPS PartialReduce kernel internals

The MIPS PartialReduce kernel follows the standard numerical computation best practices to utilize the cache usage with the *temporal* and *spatial locality*. See Algorithm 2 that uncovers the omitted details in Algorithm 1.

---

**Algorithm 2:** Detailed PartialReduce kernel for MIPS

---

**Input:** $\mathbf{Q} \in \mathbb{R}^{M \times D}$ Batch queries
**Input:** $\mathbf{X} \in \mathbb{R}^{N \times D}$ Database
**Input:** $2^W$ Bin size
**Output:** $\mathbf{V} \in \mathbb{R}^{M \times L}$ Top-$K$ values
**Output:** $\mathbf{A} \in \mathbb{N}^{M \times L}$ Top-$K$ indices

```
                    /* Block iteration over rows                              */
1   for ii ← 1 to M step ib do
                        /* Block iteration over columns                       */
2       for jj ← 1 to N step jb do
                            /* i, j, k and l are often unrolled or even vectorized  */
3           for i ← ii to ii + ib − 1 do
                                /* Starts the inner loop of the systolic arrays     */
4               yi ← 0 ;
5               for k ← 1 to D do
6                   m ← qi,k;
                                    /* Vectorized FMA (fused-multiply-add)          */
7                   for j ← jj to jj + jb − 1 do
8                       yi,j ← yi,j + m · xj,k ;
9                   end
10              end
                                /* Ends the inner loop of the systolic arrays       */
11              for j ← jj to jj + jb − 1 do
                                    /* The exact j → l mapping is determined by the compiler backend  */
12                  l ← RegisterAlignedShiftRight(j, W) ;
13                  b ← yi,j > vi,l ;                        /* COP 1: Vectorized compare */
14                  vi,l ← if b then yi,j else vi,l ;  /* COP 2: Vectorized conditional move */
15                  ai,l ← if b then j else ai,l ;     /* COP 3: Vectorized conditional move */
16              end
17          end
18      end
19  end
```

---

The *temporal locality* refers to reusing previously accessed items. In line 1, we iterate by blocks of queries. The block of queries is reused in the inner loops, achieving the temporal locality.

The *spatial locality* refers to accessing items nearby previously accessed. The block iteration loads a chunk of data points (line 2) to achieve this optimization. The same block iteration structure may apply recursively for multiple cache hierarchies till the register level.

The inner loops (indexed by $i$, $j$, and $k$ in line 3) should be unrolled or even vectorized so that every cycle can produce multiple results via the SIMD (Single Instruction Multiple Data) instructions or systolic arrays.

```python
# qy shape: f32[1024,128], db shape: f32[1048576,128]
# output shapes: f32[1024, 128], i32[1024, 128]
@jax.jit
def mips_baseline(qy, db):
  dists = jax.numpy.einsum('ik,jk->ij', qy, db)
  reshaped = jax.lax.reshape(dists, [1024, 128, 8192])
  return jax.lax.argmax(reshaped, 2, jnp.int32)
```

Listing 3: Baseline implementation without the approx_max_k operator

The algorithm principle is the same on every platform, except that the block factor and vectorization sizes are platform-dependent. We refer readers to (Golub and Van Loan, 2013) for more details.

### A.3.1 Estimate memory transfers

In Algorithm 2, memory transfer for each portion of the data is listed below:

- Query is only transferred once. Takes $4MD$ bytes.

- Database is transferred $\frac{M}{ib}$ times. Takes $4ND\frac{M}{ib}$ bytes.

- Outputs are transferred once. Takes $2 \times 4ML$ bytes.

The precise formulation for memory arithmetic intensity is

$$I_{\text{MEM}} = \frac{2MND}{4\left(MD + \frac{MND}{ib} + 2ML\right)}, \tag{20}$$

which would approach $\mathcal{O}(\min(M, N))$ as long as $L \ll \min(M, N)$ and the compiler chooses a large enough $ib$ to minimize the database transfer.

### A.3.2 Estimate COPs used

The PartialReduce kernel listed in Algorithm 1 and 2 only use $C = 3$ per dot-product. However, there are two cases that would increase the number of COPS on TPU due to the implementation constraints:

1. When the dimension $D$ is not multiple of 128, $C$ increases by 1.
2. When the database size $N$ is not power-of-2, $C$ increases by 1.

See Appendix A.5 on how it affects the real world benchmarks.

### A.3.3 Limitation of naive implementation

A naive implementation of Algorithm 1 and 2 can be composed of Reshape and ArgMax. However, the performance is not comparable to the dedicated approx_max_k operator.

Our experiment setup is as follows: let query be $\mathbf{Q} \in \mathbb{R}^{1024 \times 128}$ and database be $\mathbf{X} \in \mathbb{R}^{1048576 \times 128}$; we choose the reduction output size as $L = 128$, so the algorithm can be written as Listing 3.

We benchmark the implementations on a single-core TPU V4 instance by 100 times and collect the averaged execution time. Listing 3 took 24.9ms to compute; in comparison, our proposed new operator used in Listing 1 only took 2.6ms, which is 9.6x faster.

### A.4 Lower bound approximation of the number of bins

We care about the number of bins $L$ in the high recall region. Let the target recall $r = 1 - \epsilon$, we have

$$L \geq \frac{1}{1 - r^{1/(K-1)}} \tag{21}$$

$$= \frac{1}{1 - (1 - \epsilon)^{1/(K-1)}} \tag{22}$$

$$\approx \frac{1}{1 - \exp[\frac{\epsilon}{K-1}]} \tag{23}$$

$$= \frac{1}{1 - (1 - \frac{\epsilon}{K-1} + o(\epsilon))} \tag{24}$$

$$\approx \frac{K-1}{\epsilon}. \tag{25}$$

The approximation in (23) follows from $(1 - \epsilon)^a = (1 - \epsilon)^{\frac{1}{-\epsilon}(-\epsilon a)} \to e^{-\epsilon a}$ as $\epsilon \to 0$, and (24) follows from the Taylor expansion. $\square$

### A.5 Benchmark details

Table 2: Dataset properties and the benchmark results

|  | Glove1.2M | Sift1M |
|---|---|---|
| Dimension $D$ | 100 (Padded to 128) | 128 |
| Database size $N$ | 1,183,514 | 1,000,000 |
| Query size $M$ | 10,000 | 10,000 |
| Distance | Cosine | Euclidean |
| $C$ | 4 | 6 |
| $I_{\text{MEM}}$ | 4,758 | 4,701 |
| $I_{\text{COP}}$ | 64.0 | 42.7 |
| Measured GFLOP/s on TPU V3 | 118,524 | 118,062 |
| Measured GFLOP/s on TPU V4 | 251,166 | 172,035 |

Table 2 summarizes the dimensions and kernel properties for the two benchmarks. The memory arithmetic intensity $I_{\text{MEM}}$ is reported by the TPU profiler, and the instruction throughput intensity $I_{\text{COP}}$ is manually derived. The following show how we derive $C$ (COPs per dot-product) for each dataset.

**Glove**  The Glove dataset uses the cosine distance, which yields same search results as MIPS. As described in Appendix A.3, when the database size is not power-of-2, we pay an extra $C$ in the inner loop. Therefore the total $C$ used for the Glove benchmark are

- 3 $C$ by PartialReduce, and
- 1 $C$ by non power-of-2 database masking.

We pre-process the Glove dataset by padding the dimension from 100 to 128 to avoid one $C$. We are not bottleneck on memory bandwidth so the padding is a good trade-off for better performance.

**Sift**  The Sift dataset uses the Euclidean distance, which requires more coefficient-wise operations. In Appendix A.2 we showed that we only need to use one extra $C$ for distance computation. However, there are some other inevitable operations used in the benchmark:

- 3 $C$ by PartialReduce,
- 1 $C$ by the relaxed Euclidean distance computation,
- 1 $C$ by non power-of-2 database masking, and
- 1 $C$ by broadcasting $\frac{\|\mathbf{x}\|^2}{2}$.

Therefore the total number of $C = 6$, resulting a performance regression on TPU V4 as seen in Figure 2.

```python
import jax
from jax.experimental import maps, PartitionSpec
from jax.experimental.pjit import pjit
from functools import partial

devices = np.asarray(jax.devices())
mesh_1 = maps.Mesh(devices, ('x'))

@partial(pjit,
         in_axis_resources=(None, # replicate qy
                            PartitionSpec('x',None)),
         out_axis_resources=PartitionSpec(None, None))
def pjit_mips(qy, db):
  dot = jnp.einsum('ik,jk->ij',qy, db)
  return jax.lax.approx_max_k(dot, k=10)

with maps.Mesh(mesh_1.devices, mesh_1.axis_names):
  pjit_mips(qy, db)[1]
```

Listing 4: Multi-TPU implementation with Jax's pjit.

## A.6 Multi-TPU implementation

We demonstrate two snippets to run the approximate MIPS on multiple TPUs. Listing 4 uses the pjit (parallel just-in-time) feature of Jax. The API provides an interface for users to declare how data and computation should be sharded. In the code snippet we shard the dimension corresponds to the dataset.

Listing 5 is an alternative approach to multi-TPU approximate MIPS, and is more explicit. User must shard the data and manage the data merging manually. The program breaks down into two parts: a) parallel execution of dot-product followed by approx max $K$, and b) top-$K$ aggregation of results collected from each TPU device. Let $U$ denotes number of TPU devices, the number of items to reorder is $2KU$, where the complexity follows bitonic sort: $\mathcal{O}(KU \log^2(KU))$.

Under the hood, Listing 4 and Listing 5 compiles to identical kernels. It is not hard to see the performance scales linearly with the number of TPU used. User may choose number of TPUs according to the desired throughput and the database size.

```python
from functools import partial
import jax
import jax.numpy as jnp

# Computes approx_max_k on each TPU device in parallel.
@partial(
    jax.pmap,
    # static args: db_size, k, recall_target
    static_broadcasted_argnums=[3, 4, 5],
    out_axes=(1, 1))
def pmap_mips(qy, db, db_offset, db_size, k, recall_target):
  dists = jnp.einsum('ik,jk->ij', qy, db)
  dists, neighbors = jax.lax.approx_max_k(
      dists, k=k, recall_target=recall_target,
      reduction_input_size_override=db_size)
  return (dists, neighbors + db_offset)

# Collects partial results from each device, and
# aggregates to the final top-k.
@partial(
    jax.jit,
    static_argnames=["k"])
def mips_reorder(dists, neighbors, k):
  flat_dists = jax.lax.collapse(dists, 1, 3)
  flat_nn = jax.lax.collapse(neighbors, 1, 3)
  _, sorted_nn = jax.lax.sort([-flat_dists, flat_nn])
  return jax.lax.slice_in_dim(sorted_nn, 0, k, axis=1)

# Entry function for distributed MIPS.
def mips(qy, db, db_offset, db_size, k, recall_target):
  dists, neighbor = pmap_mips(qy, db, db_offset, db_size, k,
                              recall_target)
  return mips_reorder(dists, neighbor, k)
```

Listing 5: Multi-TPU implementation with Jax's pmap.