# OpenReview forum: "TPU-KNN: K Nearest Neighbor Search at Peak FLOP/s"
_NeurIPS.cc/2022/Conference — NeurIPS 2022 Accept_

### Official Review · Reviewer_xsza · 2022-06-28

**Rating:** 6
**Confidence:** 4
**Soundness:** 3 good
**Presentation:** 3 good
**Contribution:** 2 fair

**Summary:**

This paper studies using TPU for approximate nearest neighbor search. With careful analysis, it finds that search performance is bound by memory access and coefficient-wise operation instead of distance computation. Thus, it proposes to conduct partial reduce to reduce memory access such that search can reach peak FLOPs. Experiment results show that the proposed solution outperforms existing ones in both QPS and recall.

**Questions:**

See weakness.

**Limitations:**

Yes

**Strengths And Weaknesses:**

Strength

1.	The extended roofline analysis is interesting and shows the bottleneck of nearest neighbor search with TPU.

2.	The proposed partial reduce scheme reduces the amount of memory access and comes with theoretical analysis.

3.	The experiment results show that the proposed solution has good performance.


Weakness

1.	It would improve the paper if the authors can give a brief introduction about the characteristics of GPU and TPU. Are TPUs widely available as a commodity hardware?

2.	The proposed solution may also work for GPUs as GPUs are also limited by memory access from Table 1. Moreover, as shown in Figure 2, the proposed solution is still bound by coefficient-wise operations for Euclidean distance search. Any thoughts on reducing the number of coefficient-wise operations?

3.	The experiment study is far from the NeurIPS standard. Does the method still outperform the partial search methods (e.g., IVF) when the dataset is large (e.g., SIFT10M or 100M)? How does the proposed method perform when changing the window size W?

---

> ### Author Response · Authors · 2022-08-02
> **Response to Reviewer xsza**
>
> Thank you for carefully reviewing our paper! Please see below our responses to your comments. We use Q, W, L to denote question, weakness, and limitation correspondingly.
>
> **1. Q1**
>
> The hardware differences between TPU and GPU are very minor. See table 1 and figure 1. Dedicated GPUs and TPUs are often available through Cloud providers. For example, both GPUs and TPUs are publicly available on Google Cloud. There are also free-tier TPU resources available to researchers on request (https://sites.research.google/trc/about/).
>
> **2. Q2**
>
> > The proposed solution may also work for GPUs.
>
> The GPU implementation is on our roadmap. It is not easy because it requires reverse engineering the close sourced cuBLAS library. On TPU, naive implementation of the algorithm is 9.6x slower than the optimized version written in assembly. (Appendix A.3.3)
>
> > the proposed solution is still bound by coefficient-wise operations for Euclidean distance search. Any thoughts on reducing the number of coefficient-wise operations?
>
> See appendix A.2 for how we reduce the number of coefficient-wise operations used.
>
> **3. Q3**
>
> > Does the method still outperform the partial search methods (e.g., IVF) when the dataset is large (e.g., SIFT10M or 100M)?
>
> Thanks for the suggestion. We updated our evaluations to 7 datasets. In particular, we added a scalability benchmark (Yandex DEEP 1-10M) in Section 6.3 to address how QPS scale with the database size and number of TPUs used.
>
> > How does the proposed method perform when changing the window size $W$.
>
> Instead of looking at $W$, it is easier to estimate the performance by output size $L$ (and $W = N/L$). The program is memory bandwidth bound when
>
> $$4ML/\beta \ge 2MND/\pi.$$
>
> It follows that the window size $W$ must be greater equals to $2\pi/(D\cdot \beta)$ to keep the program operating at peak flops.
>
> ---
> We hope that our responses and the rebuttal revision can help address your concerns. If you have any other comments or feedback, please let us know! We will be happy to provide further responses. We are looking forward to hearing back from you!

---

### Official Review · Reviewer_Vrzf · 2022-07-07

**Rating:** 6
**Confidence:** 4
**Soundness:** 3 good
**Presentation:** 3 good
**Contribution:** 2 fair

**Summary:**

This paper presents a new algorithm implementation of K nearest neighbor search that is able to increase the arithmetic intensity. The distances between all queries and entries in the database are calculated in L3 BLAS fashion. Then only the top-1 entry is kept in each bin. The bin size can be adjusted based on the requirement of the recall. The roofline analysis shows a better peak throughput while the end-to-end evaluation demonstrates a better throughput-recall trade-off than the previous solutions.

**Questions:**

1. Is the proposed algorithm specialized for TPU? If it is, what special architectural support from TPU does it utilize?

2. How do you define **ONE** COP? For example, are vectorized comparisons between two 256 arrays and the comparison between two 16 arrays each counted as one COP? If not, how should I interpret the data presented in Table 1?

3. Could you provide a direct comparison of the proposed algorithm on the same hardware platform?

**Limitations:**

The authors discussed the limitation of the current implementation.

**Strengths And Weaknesses:**

## Strengths

1. This paper presents a detailed analytical model of the kernel implementation of the BLAS-based operation. The authors conclude the impact of COPs and introduce the algorithm accordingly.

2. The proposed solution is simple yet effective. The adjustment of the bin size provides a simple method to balance recall and throughput.

## Weaknesses

1. **Unclear hardware motivation.** The paper is entitled 'TPU-KNN'. However, throughout this paper, I'm not able to find any information indicating that the proposed algorithm requires any special architectural support from TPU rather than other accelerators. It seems that the same analysis could also apply to GPU or other accelerators.

2. **Vague definition of COPs.** I cannot understand the usage of the concept of coefficient-wise operations (COPs). In Table 1, comparing with the datasheet of A100/V100, I can understand COPs as a FLOP in the vector cores (or SMs in NV's terminology) while the FLOP means a half-precision FLOP in the tensor core. However, in the later description like Algorithm 1, one COP seems to be a general non-matrix operation without considering the problem size. In this case, one COP, for example, vectorized comparison, could include multiple FLOPs and seems to be unmatched by what is presented in Table 1.

3. **Incomplete Evaluation.** Based on what I've mentioned in 1, the evaluation part lacks a fair baseline. I can understand the advantage of recall of the proposed algorithm. However, it is hard to understand whether the performance gain over GPU comes from the algorithm itself or the higher efficiency of TPU. To isolate this factor, the authors should either provide the result of the proposed algorithm implemented on GPUs or the baseline algorithms implemented on TPU.

---

> ### Author Response · Authors · 2022-08-02
> **Response to Reviewer Vrzf**
>
> Thank you for carefully reviewing our paper! Please see below our responses to your comments. We use Q, W, L to denote question, weakness, and limitation correspondingly.
>
> **1. Q1**
>
> Our algorithm is generic and should achieve peak performance on all platforms. Nevertheless, implementing an optimized kernel is not trivial so we only deliver the TPU implementation for this submission. The challenges are as follows:
> 1. A fused implementation will likely involve integrating efficient CPU/GPU matrix multiplication implementation with customized reduce logic and can be quite challenging. Some matrix multiplication libraries are complex and often closed sourced e.g. cuBlas on GPU and MKL on CPU.
> 2. A naive, non-fused implementation can work across platforms (Listing 3 of Appendix). On TPU, the naive implementation of the algorithm is 9.6x slower than the optimized version written in assembly. (Appendix A.3.3 in the updated revision)
>
> **2. Q2 and W2**
>
> On nvidia GPUs, COPs correspond to CUDA cores (general vectorized ops like CPU SIMD instructions), and FLOPs correspond to Tensor cores (systolic arrays which can only perform matrix multiplications). For both TPU and GPU (in newer generations like V100, A100, and H100), performing matrix multiplication with systolic arrays (tensor core) is faster than just using the vector instructions. In this paper we show that an algorithm that uses both systolic arrays and vector instructions can be bound by the vector instruction throughput.
>
> Following your example, performing arithmetic between two 256 arrays of F32, yields 256 COPs (per cycle) for this SIMD operation. On all platforms, COP/s is defined as SIMD vector length * number of cores * chip frequency.
>
> **3. Q3 and W3**
>
> We achieved the efficiency on TPU by writing TPU assembly code and ensuring critical loops all happen in registers. While an unoptimized implementation should be straightforward and work across platforms (Listing 3 of Appendix), it performed much slower than optimized version on TPU (9.6x, Appendix A.3.3). To implement similar optimization on GPU requires significant effort, possibly with inner details of cuBlas. It is on our roadmap but It will take much longer than the allowed rebuttal period.
>
> **4. W1**
>
> The reviewer is correct that the analysis is not limited to TPU-ANN algorithms, and can be extended to other platforms (CPU, GPU, TPU or even FPGA), as well as other algorithms that can also be bottlenecked by memory bandwidth or COPs. In this paper we choose to focus on TPU-ANN only.
>
> ---
> We hope that our responses and the rebuttal revision can help address your concerns. If you have any other comments or feedback, please let us know! We will be happy to provide further responses. We are looking forward to hearing back from you!

---

> > ### Comment · Reviewer_Vrzf · 2022-08-07
> > **Possible for a TPU baseline?**
> >
> > Dear authors,
> >
> > Thanks for your response! Most of my concerns are addressed.
> >
> > One last question is that, to understand the contribution of the proposed algorithm, is it possible to provide a baseline result (e.g, Flat or PQ)? Would that require extensive effort for optimization?

---

> > > ### Author Response · Authors · 2022-08-08
> > > **Response to Reviewer Vrzf**
> > >
> > > Thank you for your reply.
> > >
> > > 1. Flat (dot-product + exact TPU TopK) is memory bandwidth bound (minor) + TopK instruction bound (huge). The speed difference varies by k. We use a smaller query size to verify the speed difference to avoid OOM. The benchmark is evaluated on Glove-100 (1M datapoints), TPU-V4 (single core), with query size 2048.
> > > * Exact top-1: 33.6 ms
> > > * Exact top-10: 85.6 ms
> > > * Exact top-100: 606 ms
> > > * Our approx top-1: 5.57 ms
> > > * Our approx top-10: 6.01 ms
> > > * Our approx top-100: 11.5 ms
> > >
> > > See also Section 4.3 for why performing exact TopK is slow (instruction bound.)
> > >
> > > I'll try to squeeze the flat baseline benchmark result into the paper.
> > >
> > > 2.1. PQ is a great algorithm to save up memory bandwidth but is difficult to implement efficiently on accelerators. On CPU some PQ implementations (Google scann and Facebook faiss) uses X86 vpshufb instruction to perform quantized lookup in 1 cycle. On GPU/TPU the lookup is much more expensive. In Figure 3 you can find Faiss IVF-PQ is significantly slower than pure IVF versions.
> > >
> > > 2.2. From 1. we can see that TopK itself is instruction bound, therefore adding a ANN solution that saves up memory bandwidth but increases instruction used would only make the program slower.
> > >
> > > Let us know if it addresses your concern.

---

> > > > ### Comment · Reviewer_Vrzf · 2022-08-08
> > > > **Reply**
> > > >
> > > > That makes sense. I've revised my rating.
> > > >
> > > > Thank you!

---

### Official Review · Reviewer_S1Tw · 2022-07-11

**Rating:** 5
**Confidence:** 5
**Soundness:** 2 fair
**Presentation:** 4 excellent
**Contribution:** 2 fair

**Summary:**

This paper presents an ANN algorithm on TPU and analyzes the memory and instruction bandwidth of ANN algorithms.


**Questions:**

1. Could we have other methods on GPUs/TPUs, e.g., hashing and graph-based methods besides the FAISS baseline, on more measures, e.g. cosine?
3. Could we show that the proposed algorithm can achieve high recalls for most ANN datasets?

**Limitations:**

I suggest the authors to discuss the limitation of the algorithm, e.g., how to adapt the problem to other common similarity measures.

**Strengths And Weaknesses:**

Strong Points
----
1. The problem is well-motivated.
2. The related work is comprehensively studied and discussed. And the entire story is easy to follow for readers.
3. The methodology to study the algorithm from the hardware bottlenecks is interesting.

Weak Points
----
1. I suggest the authors to include a brief discussion of TPU, e.g., what it can be done and what it can not be done (efficiently).
2. The experimental evaluation is a bit problematic. How about other methods on GPUs/TPUs, e.g., hashing and graph-based methods besides the FAISS baseline?
3. In Figure 3, the highest recall shown for Glove is 0.9. Is there a recall limitation for the proposed algorithm?
4. The technical contribution of the bi-stage partial reduction and scoring is limited.

---

> ### Author Response · Authors · 2022-08-02
> **Response to Reviewer S1Tw**
>
> Thank you for carefully reviewing our paper! Please see below our responses to your comments. We use Q, W, L to denote question, weakness, and limitation correspondingly.
>
> **1. Q1, W2, and L1**
>
> We choose to show FAISS in our graph since it is a representative CPU based method. There have been comprehensive benchmarks comparing hashing or graph based methods to FAISS baselines on datasets such as Glove / SIFT (http://ann-benchmarks.com/). In most cases, FAISS performs competitively. Cosine distance search is equivalent to maximal inner product search (MIPS) on normalized vectors (Ln 49). In the updated revision, the following datasets are evaluated with the cosine distances: Glove, Nytimes, Last.fm, and Yandex Deep.
>
> **2. Q2**
>
> We updated our evaluations to use 7 datasets in Section 6, and achieved high recalls on all datasets.
>
> **3. W1**
>
> TPU and GPU have very little differences in hardware capabilities (see table 1 and figure 1). Some people have a misconception from the internet that TPUs only work well for neural networks, and this paper is an example that counters the misconception. The classical linear algebra solvers (qr, svd, choleskey, etc.) in jax.numpy.linalg are other examples that work well on TPU, but are not neural networks.
>
> **4. W3**
>
> In Figure 3. (Glove 1.2M, D=100), the highest recall of "Faiss-{A100|V100} ivf,flat" is 0.90017. "TPU {V3|V4}-Ours" has higher recall numbers (0.97736-0.98055).
>
> **5. W4**
>
> We simplify the pseudo code in Algorithm 1 for the ease of analysis. Nevertheless, the implementation is not trivial. The proposed solution written in TPU assembly is 9.6x faster than a naive implementation of the algorithm (Appendix A.3.3. in the updated revision)
>
> ---
> We hope that our responses and the rebuttal revision can help address your concerns. If you have any other comments or feedback, please let us know! We will be happy to provide further responses. We are looking forward to hearing back from you!

---

### Official Review · Reviewer_sFRW · 2022-07-11

**Rating:** 6
**Confidence:** 4
**Soundness:** 3 good
**Presentation:** 3 good
**Contribution:** 3 good

**Summary:**

The paper presents a new NN-search algorithm, that reaches the peak performance on TPUs. The paper is based on observations of hardware architectural properties and the so called roofline performance model. The algorithm is implemented for TPUs. The evaluation is done on two TPU versions using two KNN datasets, and compared to several GPU algorithms.


**Questions:**

* The algorithm descriptions (Algorithm 1 and Algorithm 2 (suppl. mtrl)) looks relatively clear and straight-forward to implement on other platforms. Can you elaborate a bit on why it would be such a substantial effort to do?


**Limitations:**

I think the authors have adequately addressed the limitations of their work.


**Strengths And Weaknesses:**

Strengths
+ Nice connection between theoretical observations and practical results
+ Good performance
+ Can have significant practical impact

Weaknesses
- Limited evaluation, only evaluated on two versions on TPUs, and two datasets
- Would have been interesting to see how general the algorithm is, i.e., would it reach the same performance limits in other hardware platforms also?

---

> ### Author Response · Authors · 2022-08-02
> **Response to Reviewer sFRW**
>
> Thank you for carefully reviewing our paper! Please see below our responses to your comments. We use Q, W, L to denote question, weakness, and limitation correspondingly.
>
> **1. Q1 and W2**
>
> While the algorithm itself isn't overly complex, implementing an optimized kernel isn't trivial.
> 1. A fused implementation will likely involve integrating efficient CPU/GPU matrix multiplication implementation with customized reduce logic and can be quite challenging. Some matrix multiplication libraries are complex and often closed sourced e.g. cuBlas on GPU and MKL on CPU.
> 2. A naive, non-fused implementation can work across platforms (Listing 3 of Appendix). On TPU, the naive implementation of the algorithm is 9.6x slower than the optimized version written in assembly. (Appendix A.3.3 in the updated revision)
>
> **2. W1**
>
> We updated our evaluations to use 7 datasets in Section 6.
>
> ---
> We hope that our responses and the rebuttal revision can help address your concerns. If you have any other comments or feedback, please let us know! We will be happy to provide further responses. We are looking forward to hearing back from you!

---

> > ### Comment · Reviewer_sFRW · 2022-08-08
> > **Reply**
> >
> > Dear authors,
> >
> > Thanks for your response and answers. Good that you have updated the evaluation with more results. I understand the challenges with implementing optimized versions of your algorithm on different platforms.

---

### Meta-Review · Area_Chair_tckz · 2022-08-25

**Recommendation:** Accept
**Confidence:** Certain

**Metareview:**

The authors design an efficient implementation of nearest neighbor search on a TPU accelerator unit. The implementation is motivated by a refined roofline performance model that takes into account the memory and instruction bottlenecks that are found to be significant and that are not typically optimized for. Empirical results demonstrate that the proposed TPU solver outperforms state-of-the art GPU solvers. The package is available on Tensorflow.

The reviewers agree that the paper is well written, well structured and the proposed method can have significant practical impact.
Some concerns regarding the evaluation came up in the reviews but the additional experiments provided could address most of these concerns. What remains is a question on whether the performance gain over GPU comes from the algorithmic optimization itself or the higher efficiency of the TPU, and whether the algorithmic optimization would be similarly effective on other accelerators. The reviewers agree that performing such an analysis is outside the scope of this work. However, I want to encourage the authors to incorporate additional discussion to help the reader understand what parts of the work are specific to TPUs.

Overall this paper represents a well executed piece of work at the intersection between algorithm design and systems with an open source package that is available to the community. I recommend acceptance.


**Award:**

No

---

### Decision · Program_Chairs · 2022-09-14

Accept